# A pandemic-enabled comparison of discovery platforms demonstrates a naïve antibody library can match the best immune-sourced antibodies

Fortunato Ferrara[1], M. Frank Erasmus[1], Sara D'Angelo[1], Camila Leal-Lopes[2], André A. Teixeira[2], Alok Choudhary[3], William Honnen[3], David Calianese[3], Deli Huang[4], Linghan Peng[5], James E. Voss[5], David Nemazee[5], Dennis R. Burton[5,6], Abraham Pinter[3] & Andrew R. M. Bradbury[1✉]

As a result of the SARS-CoV-2 pandemic numerous scientific groups have generated antibodies against a single target: the CoV-2 spike antigen. This has provided an unprecedented opportunity to compare the efficacy of different methods and the specificities and qualities of the antibodies generated by those methods. Generally, the most potent neutralizing antibodies have been generated from convalescent patients and immunized animals, with non-immune phage libraries usually yielding significantly less potent antibodies. Here, we show that it is possible to generate ultra-potent ($IC_{50} < 2$ ng/ml) human neutralizing antibodies directly from a unique semisynthetic naïve antibody library format with affinities, developability properties and neutralization activities comparable to the best from hyperimmune sources. This demonstrates that appropriately designed and constructed naïve antibody libraries can effectively compete with immunization to directly provide therapeutic antibodies against a viral pathogen, without the need for immune sources or downstream optimization.

[1] Specifica Inc, Santa Fe, NM 87505, USA. [2] Bioscience Division, New Mexico Consortium, Los Alamos, NM 87544, USA. [3] Public Health Research Institute, New Jersey Medical School, Rutgers, The State University of New Jersey, Newark, NJ 07103, USA. [4] Life Sciences Institute, Zhejiang University, Hangzhou, China. [5] Department of Immunology and Microbiology, The Scripps Research Institute, La Jolla, CA 92037, USA. [6] Ragon Institute of MGH, MIT and Harvard, Cambridge, MA 02139, USA. ✉email: abradbury@specifica.bio

As of early November 2021, over 248 million people worldwide have been documented to be infected with SARS-CoV-2, the novel coronavirus causing COVID-19, resulting in over 5 million deaths. This has led to worldwide interest in the rapid development of effective diagnostic, prophylactic, and therapeutic options for this pandemic. Several Covid-19 vaccine candidates have been approved and are being broadly distributed[1–3], but it is unlikely the bulk of the world's population will be vaccinated before 2023, during which time the virus will continue to spread and mutate, risking the development of vaccine-resistant variants. On the therapeutic front, in addition to drug repurposing[4] there has been intense interest in generating neutralizing antibodies against SARS-CoV-2[5]. Most biotech/pharma companies and academic laboratories with antibody technology have generated antibodies against this virus, concentrating on the spike (S) protein, a homotrimeric glycoprotein anchored in the viral membrane, responsible for binding to angiotensin-converting enzyme-2 (ACE2)[6], the host cell receptor mediating viral entry.

Never in antibody history have so many different groups attempted to isolate antibodies against a single target. These concerted worldwide efforts represent a unique opportunity to compare different methods of antibody generation for their efficiency and speed, and the properties of the derived antibodies for their affinity, neutralizing potency and developability. The most sensitive CoV-2 neutralization target is the receptor-binding domain (RBD) of the S protein, and effective antibodies against this site block the binding of the virus to its receptor, thereby inhibiting entry. Initial attempts to obtain neutralizing antibodies consisted in analyzing the cross-neutralizing activity of mAbs specific for SARS-CoV-1 from the 2003 outbreak[7,8]. Only a small minority of such antibodies bound the SARS-Cov-2 spike protein or neutralized the virus, and these possessed relatively low potencies[7,8]. Simultaneously many groups focused on antibodies obtained from the B-cells of convalescent COVID-19 patients[9–23] or immunized transgenic mice[13] yielding multiple antibodies with potent SARS-CoV-2 neutralization properties. In parallel, many groups selected antibodies from preexisting naïve in vitro libraries, a technology first described 30 years ago[24], generally yielding SARS-CoV-2 spike antibodies with lower affinities and potencies[25–32], suggesting that naïve libraries are unable to generate antibodies matching those from immune sources.

We recently developed a naïve semisynthetic library platform[33,34] that has routinely generated highly diverse, drug-like antibodies with subnanomolar affinities. The enhanced performance of the platform arises from its design, which involves embedding whole CDRs from natural B cell receptors into well-behaved clinical antibody scaffolds. We reasoned that the use of well-behaved clinical scaffolds would ensure stable VH/VL pairing, while the stringent quality control applied to antibodies undergoing natural B-cell maturation would equally apply to their individual CDRs, ensuring that natural CDRs would fold correctly when placed within the appropriate germline framework. The HCDR3s were directly amplified from the mRNA of B cells purified from the LeukoPaks of ten donors, while the remaining "naturally replicated" CDRs were identified from next-generation sequencing (NGS) of 40 donors and synthesized on arrays after eliminating those containing sequence liabilities, speculating that a library in which the replicated natural CDRs were informatically purged of sequence liabilities would provide antibodies with superior biophysical properties. As all the CDRs in such a library are derived from natural sources, diversity arises from recombination between, but not within, CDRs. While antibodies derived from this library platform have been shown to have remarkable properties against a number of different targets[33,34], it has not been possible to directly compare them to those derived from other naïve libraries, or antibodies generated from immune sources against the same target. The advent of the pandemic and corresponding worldwide focus on the SARS-CoV-2 spike protein has now allowed us to do so.

Here, we describe a panel of antibodies against the SARS-CoV-2 spike protein directly selected from this naïve library platform, with neutralizing IC50s comparable to the best obtained with more time-consuming strategies dependent on immune B cells, demonstrating that naïve libraries of this kind are a viable alternative to immune cells as antibody sources for viral pathogens and possibly additional targets as well.

## Results

**Anti-SARS-CoV-2- antibodies identification and initial validation.** We adapted our selection pipeline, summarized in Fig. 1a, to the generation of human single-chain Fv (scFv) antibodies against the S1 domain of the SARS-CoV-2 spike protein. Biotinylated SARS-CoV-2 spike protein S1 was used to select antibodies from the semi-synthetic phage scFv antibody library described above[33,34], followed by yeast display sorting. This combines the advantages of phage display (extremely high diversity starting library) with those of yeast display (precise selection calibration by flow cytometry)[35,36]. Before subcloning into the yeast display vector, a strong polyclonal signal (Fig. 1b) was detected against the S1 domain as well as the RBD. The population subcloned and displayed on yeast was further enriched for binders specific for the S1 domain (Supplementary Fig. 1 explains the gating and sorting strategy). After three rounds of yeast sorting, with decreasing amounts of antigen at each cycle, we were able to detect a large scFv-yeast-displaying binding population by flow cytometry when the cells were stained with 1 nM of S1. Staining the same population with RBD showed that the majority of enriched antibodies were directed towards the RBD portion of the protein (Fig. 1c).

**Batch reformatting of scFv populations to full-length IgG.** The output obtained after the final yeast sorting step was reformatted as full-length IgGs using a methodology based on that described by Xiao et al.[37], in which IgG expression is driven by a single bicistronic vector that retains the original VH/VL pairing of the selected scFv (Supplementary Fig. 2 and materials and methods). 96 bacterial colonies from the final selected population were Sanger sequenced, providing 31 unique antibody sequences. When the amino acid sequences of the 31 antibodies were compared to a comprehensive database (CoV-AbDab) of antibody sequences selected against SARS-CoV-2[38], the HCDR3s length distribution was similar to the other antibodies previously described (Supplementary Fig. 3a) with slightly better profiles for the numbers of charged amino acids (Supplementary Fig. 3b) and Parker hydropathy profiles (positive values indicate greater hydrophilicity) (Supplementary Fig. 3c). Moreover, 100% of the HCDR1-2 and LCDR1-3 sequences of the selected antibodies lacked 15 different sequence liability motifs (Supplementary Table 1 and Supplementary Fig. 3g), while 84% of the HCDR3s contained zero or only a single (isomerization) sequence liability (Supplementary Fig. 3h), reflecting the deliberate prior removal of sequence liabilities from all CDRs, except HCDR3, in the library. This contrasts with the CDRs of antibodies in CoV-AbDab, for which 20% lacked sequence liabilities in HCDR1-2 and LCDR1-3, and only 35% contained zero or a single sequence liability in HCDR3. Casirivimab and Imdevimab, two anti-SARS-CoV-2 antibodies with emergency approval, have one and three sequence liabilities, respectively. Finally, we quantified the total number of CDRs per antibody that contain at least 1 liability, with a potential maximum value of six (indicating all six CDRS have at

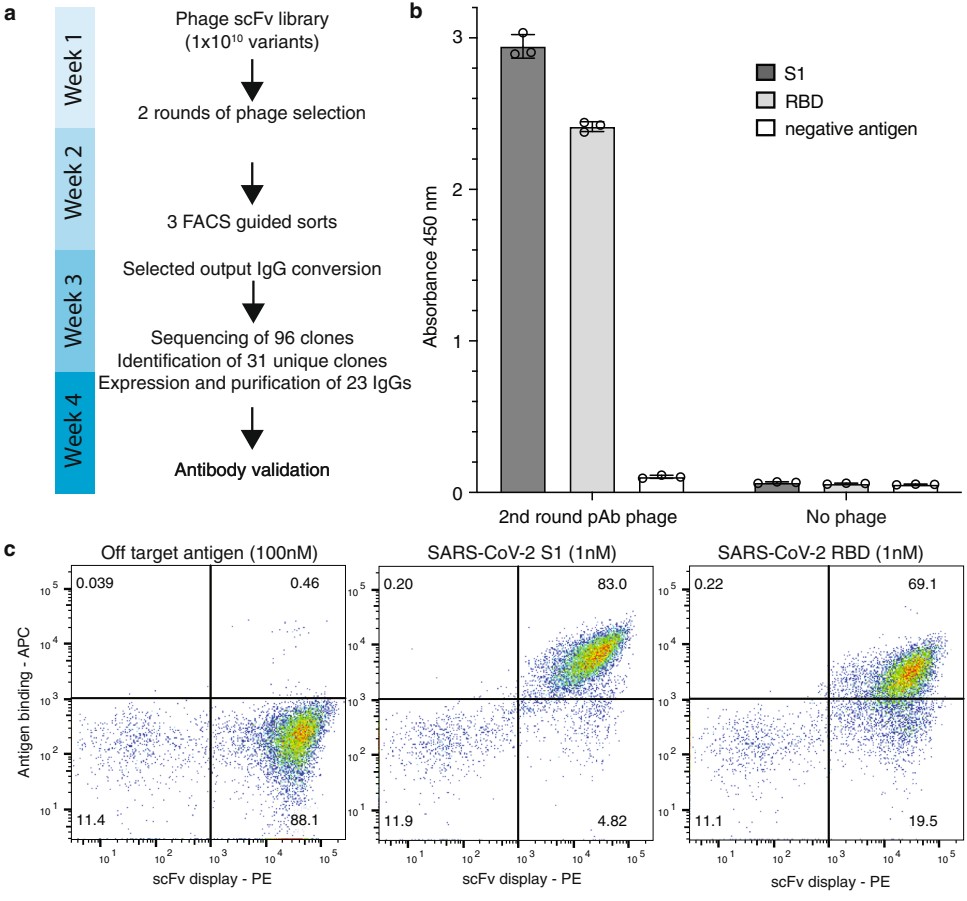

**Fig. 1 Antibody selection. a** Timeline of the entire pipeline: from phage selection to the validation of the selected antibodies. **b** Binding signal of the polyclonal phage population obtained after two round of phage display selections against SARS-CoV-2 S1 and SARS-CoV-2 RBD. Data represent the average of three polyclonal phage preparations ($n = 3$) +/− SD. Source data are provided as a Source Data File. **c** Analysis of binding signals by flow cytometry of the final antibody population obtained after yeast display against SARS-CoV-2 S1 and SARS-CoV-2 RBD.

least one liability (Supplementary Fig. 3i)). This revealed that almost all antibodies have zero or only one CDR with a sequence liability, while 67% of antibodies in CoV-AbDab have liabilities in two or more CDRs, and only 35% contain zero to one CDR with a sequence liability.

Twenty-three of 31 antibodies (sequences are reported in Supplementary Table 2) were taken forwards for characterization, since eight antibodies did not yield sufficient immunoglobulin to perform the designed experiments.

**Antibody affinity measurements.** An array SPR instrument (Carterra LSA) was used to determine the binding kinetics of the IgG antibodies selected from the semisynthetic library platform, together with five previously described antibodies obtained from convalescent patients: CC6.29 and CC6.30 are highly potent neutralizing immune antibodies specific for the most commonly recognized RBD-A epitope, CC12.17 and CC12.18 recognize the different RBD-B epitope with lower affinities and neutralizing activities[18], and CR3022 is an anti-SARS-CoV-1 RBD antibody also recognizing SARS-CoV-2 RBD (RBD-C) with low affinities and neutralization activity[7,18]. Moreover, four antibodies obtained from previously published in vitro "naïve" phage libraries were also produced and used as comparisons: Ab1[39], STE73-2E9[40], RBD1, and its affinity matured version mRBD1.5[41] (naïve antibodies), were chosen based on their higher reported affinities and sequence availability.

Antibodies were captured in a ligand array and kinetic rates and affinity (or avidity for the trimeric spike) constants

determined for the RBD or trimer[42]. The sensorgrams (Supplementary Fig. 4a, b) for the selected antibodies show affinities ranging from ≤34 pM to 6.3 nM (median 295 pM) for the RBD, and avidities from 38 pM to 32.7 nM (median 174 pM) for the trimer, while the affinities of CC6.30 and CC6.29 were 1.69 nM and 742 pM for RBD and 242 pM and 399 pM for the trimer. CC12.17, CC12.18, and CR3022 had affinities of 84.6 nM, 2.2 nM, and 11 nM, respectively, for the RBD (Fig. 2a–c), in line with previously published results[7,18]. For Ab1, STE73-2E9 and mRBD1.5 affinities for the RBD were measured as 47.0 nM, 15.0 nM and 69.0 nM respectively. These values are significantly worse than those previously published. No binding activity was observed for RBD1 (Fig. 2a–c), which was subsequently explained by the presence of a glycosylation site, which would have been glycosylated when expressed in mammalian cells as described here, but not in the original bacterial expression. This site is lost in the affinity matured variant, mRBD1.5, the published affinity of which was measured in the Fab, and not IgG, format.

The affinities of the selected antibodies with higher neutralization potencies (A12, F1, C2, #41, and C4) ranged from 52 to 339 pM for the RBD, with all but A12 showing higher avidity for the spike trimer. Most of the other antibodies also bound more tightly to the trimer than the RBD, but some (G5, A12, A11, C7, and the CC12.18 antibody) showed the opposite effect (Fig. 2b–d).

**Binning characterization.** High-throughput epitope binning was also analyzed using the SPR array, with each individual antibody

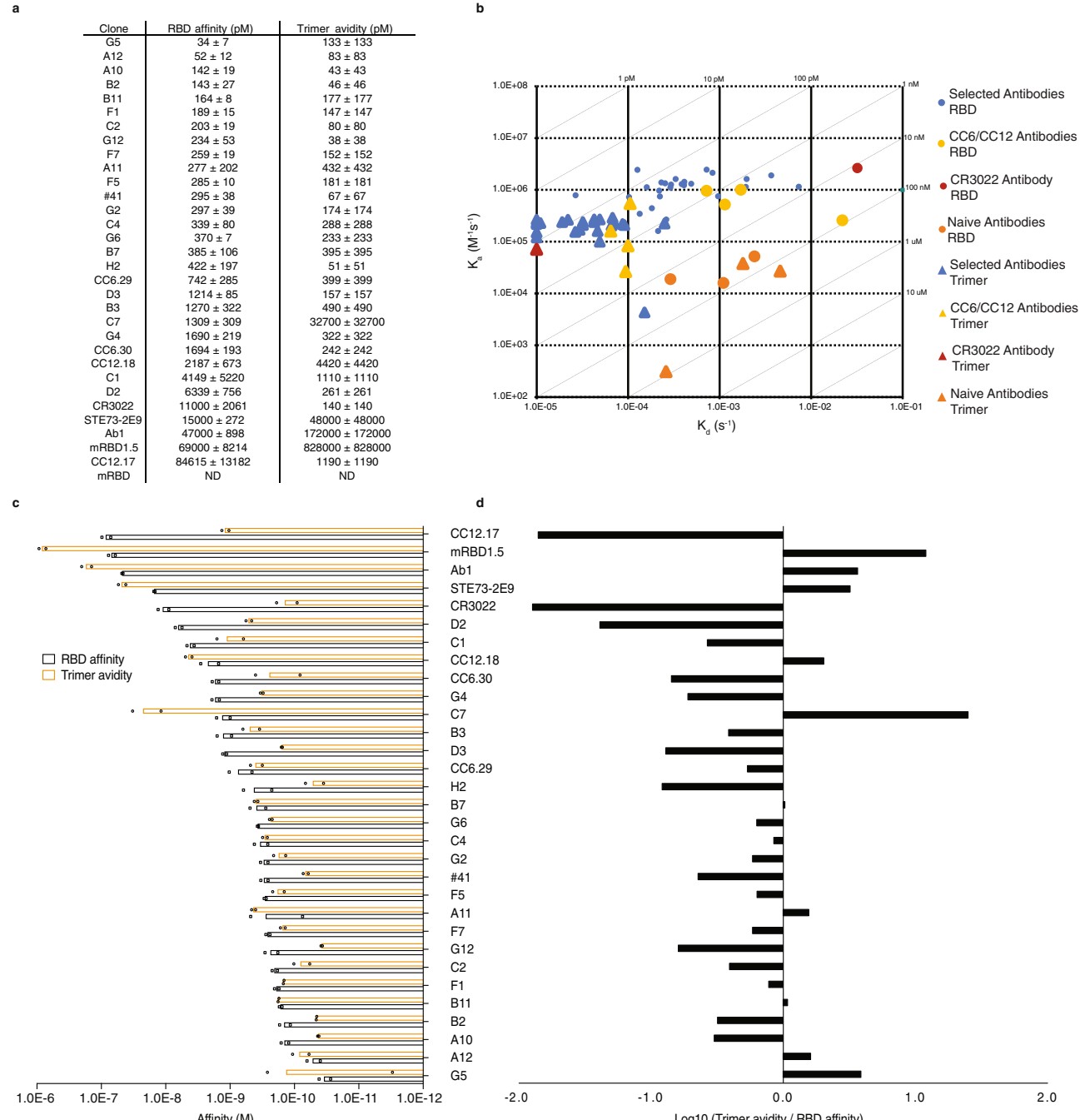

**Fig. 2 Kinetic characterization of selected antibodies. a** Measured affinities and avidities of the selected antibodies (IgGs) against SARS-CoV-2 RBD and trimer respectively. Data represent the average of duplicates of independent affinity measurements for each of the tested antibodies ($n = 32$) $+/-$ SD. **b** Isoaffinity plot of the antibodies against RBD (circle) and trimer (triangle) plotting off-rate (s$^{-1}$) against on-rate (M$^{-1}$s$^{-1}$). Data represent the average of duplicates of independent affinity measurements for each of the tested antibodies ($n = 32$). **c** Affinity histogram ranked on RBD affinities plotted with corresponding trimer avidities. Data represent average of duplicates of independent affinity measurements for each of the tested antibodies ($n = 32$) $+/-$ SD. **d** Ratio of the trimer avidity values divided by the RBD affinity. Source data are provided as a Source Data File.

covalently immobilized to the surface of the chip ("ligand"), and antigen and competitive antibodies ("analytes") individually bound to the ligands. All antibodies were tested as both analytes and ligands and compared to the 5 previously described mAbs, defining three epitope bins[7,18]. Most selected antibodies binned with the CC6.29/30 antibodies (RBD-A) and sandwiched with the CC12.17/18 (RBD-B) and CR3022 (RBD-C) antibodies (Fig. 3 and Supplementary Fig. 6a–c). However, #41 and B3 showed a distinct binning profile, sandwiching with CC12.17 and CR3022,

but not CC12.18 (Supplementary Fig. 6d), suggesting binding of a similar epitope and/or allosteric effects for #41 and B3. The remaining selected antibodies (Supplementary Fig. 5e) sandwiched with CR3022 and the CC12.17/18 set, while completely blocking all remaining antibodies.

**SARS-CoV-2 neutralization by the selected antibodies.** The antibodies were tested in a previously described lentivirus pseudovirus neutralization assay[18] (Table 1) with IC$_{50}$s ranging from

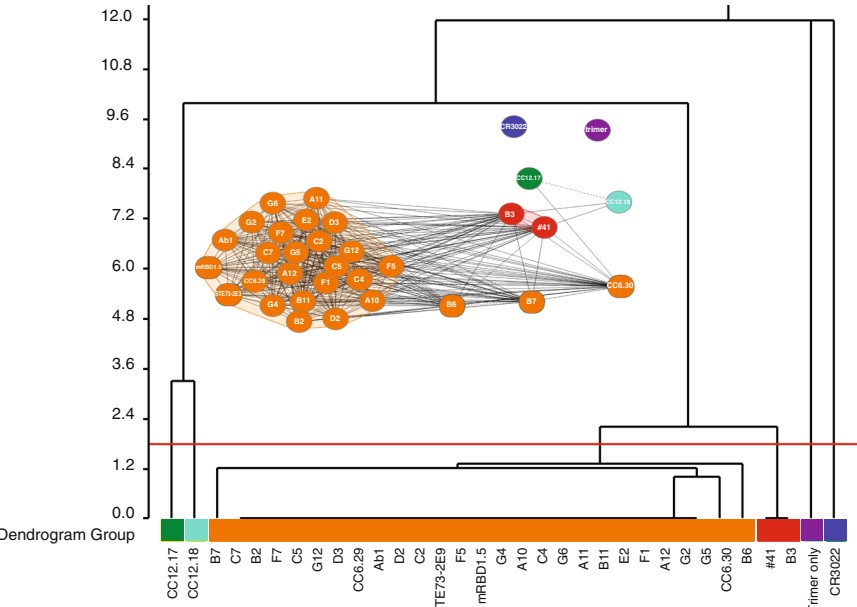

**Fig. 3 Binning of discovered leads.** Advanced binary dendrogram of each of the antibodies in a pairwise manner. Most of the selected antibodies belong to the RBD-A epitope group, reported as the one with highest neutralization potency.

**Table 1 Neutralization potencies of the selected antibodies obtained from the pseudovirus assay.**

| Clone | SARS-CoV-2 pseudovirus IC50 (ng/mL) |
|---|---|
| #41 | 3.7 |
| A10 | 27.2 |
| A11 | 22.3 |
| A12 | 1.7 |
| B2 | 85.2 |
| B3 | 3.5 |
| B7 | 10.1 |
| B11 | 12.2 |
| C1 | 48.6 |
| C2 | 41.3 |
| C4 | 8.1 |
| C7 | 9.1 |
| D2 | 13.4 |
| D3 | 1.8 |
| F1 | 5.2 |
| F5 | 2.6 |
| F7 | 6.7 |
| G2 | 6.6 |
| G4 | 20.2 |
| G5 | 53.9 |
| G6 | 4.4 |
| G12 | 5.3 |
| H2 | 4.0 |
| CC6.29 | 48.6 |
| CC6.30 | 6.4 |
| Ab1 | 27.8 |
| STE73-2E9 | 9.2 |

1.7 to 85.2 ng/mL. Eleven of the antibodies selected here were more potent than CC6.29 and CC6.30 (IC$_{50}$s 48.6 and 6.4 ng/mL) when directly compared in the same assay, and more potent than the antibodies generated from other naïve recombinant antibody libraries, among which we were able to measure IC$_{50}$s for only two: Ab1 (IC$_{50}$ = 27.8 ng/mL) and STE73-2E (IC$_{50}$ = 9.2 ng/mL). Figure 4a shows the neutralization profiles of a subset of ten antibodies, chosen for their binning and affinity properties.

Testing the same ten antibodies in a live viral neutralization test using Vero E6 cells infected with pathogenic SARS-CoV-2 (Fig. 4b), showed that the selected antibodies also have ultra-potent neutralizing activity against infectious SARS-CoV-2, with four antibodies demonstrating IC$_{50}$s < 2 ng/ml (Table 2), and the best an IC$_{50}$ of 1.3 ng/ml. IC$_{50}$s in the live virus assay correlated with competition for ACE2 receptor binding to RBD (Fig. 4c), as well as the pseudoviral neutralization assay, as previously reported[18].

We also assessed these ten antibodies for pseudoviral neutralization activity against the SARS-CoV-2 Alpha (B.1.17) and Beta (B.1.35) strains, together with CC6.29, CC6.30, Ab1, and STE73-2E9. Both variants conferred some resistance to neutralization by antibodies B3, as well as antibodies D2 and D3, which have the lowest RBD affinities. Remarkably, the remaining antibodies retained the neutralization potencies of the wild-type virus, while all the previously described antibodies isolated from convalescent patients or from other recombinant antibody libraries either had much lower neutralization activity against the beta strain (CC6.29) or no activity at all (Fig. 4d and Table 3).

**Developability properties.** Next, we assessed the developability of the antibodies validated for live virus neutralization activity by conducting a series of assays to measure their experimentally-derived biophysical properties. In particular, we analyzed levels of polyreactivity, self-interaction, thermal stability, and long-term aggregation propensity by employing SPR against panel of poly-specificity reagents[43], affinity-capture self-interaction nanoparticle spectroscopy (AC-SINS)[44], size-exclusion chromatography (SEC) after prolonged exposure to high temperature, 37 °C, and melting temperature (Tm) using differential scanning fluorimetry (DSF)[45,46], respectively. The aim of these studies were to provide insight into how well the selected antibodies perform relative to the single clinical antibody scaffold (Abrilumab) from which they were derived and to a poorly developable antibody (Sirukumab) that failed in the advanced stage of clinical trials[47]. We also included a polyspecific antibody (control mAb) in the SPR assay to provide a positive control for polyreactive binding.

All of the selected antibodies tested show favorable developability profiles with a) values similar to the clinical antibody, b)

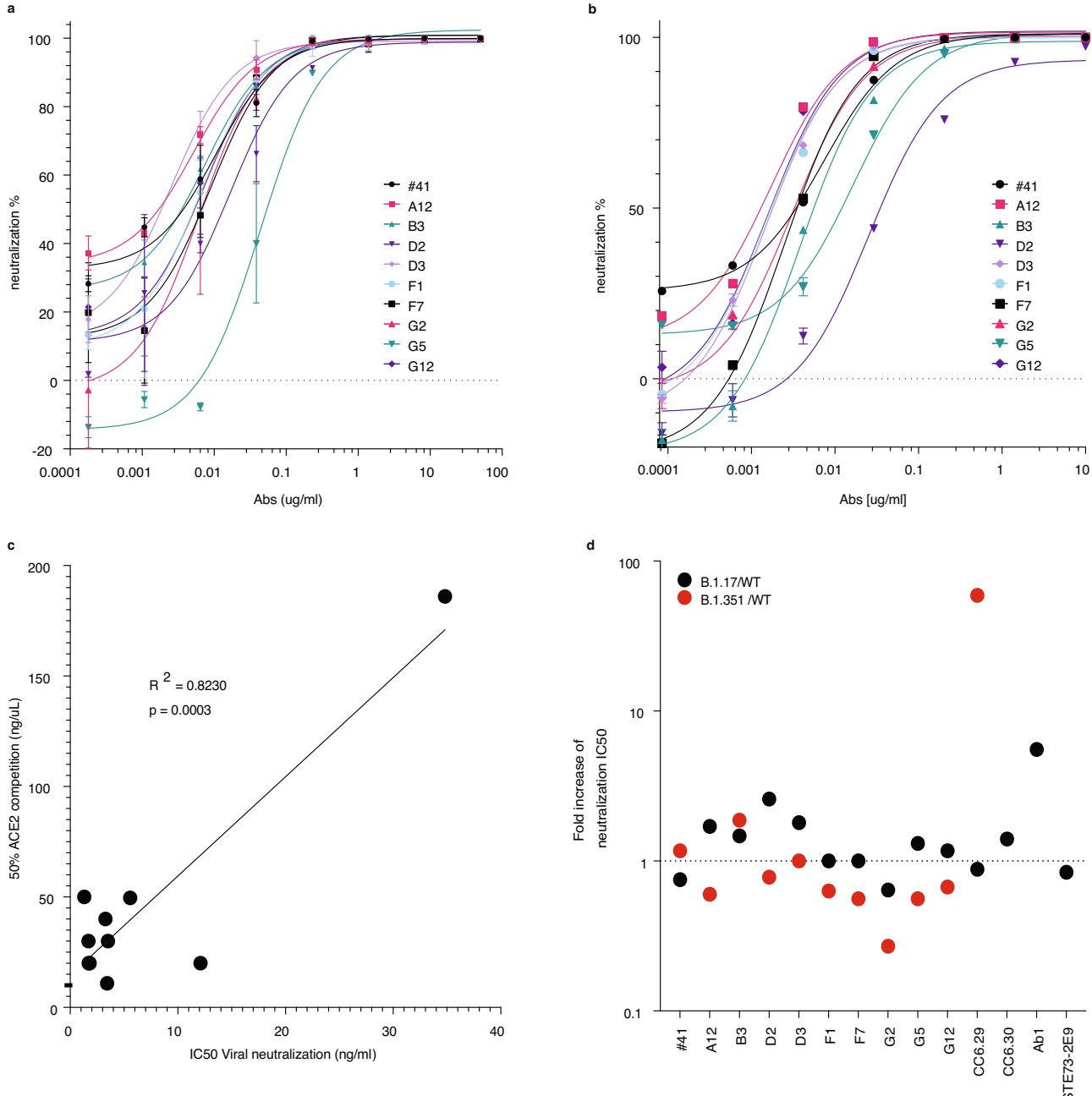

**Fig. 4 SARS-CoV-2 neutralization. a** SARS-CoV-2 pseudovirus neutralization by the selected antibodies. Data represent the average of triplicates of independent pseudovirus neutralization measurements of the different antibodies ($n = 10$) $+/-$ SD. **b** Live SARS-CoV-2 virus infection of Vero cells neutralization by the selected antibodies. Data represent the average of triplicates of independent live virus neutralization measurements of the selected antibodies ($n = 10$) $+/-$ SD. **c** The neutralization potencies (IC$_{50}$) on the SARS-CoV-2 virus were correlated with the measure of competition to ACE-2 receptor. Pearson's correlation coefficient and two-tailed test p-value are indicated. **d** Ratio of the neutralization potencies of the selected antibodies against B.1.1.7 (Alpha) and B.1.135 (Beta) variants versus wild-type SARS-CoV-2. Source data are provided as a Source Data File.

improved developability profiles relative to a polyspecific mAb (control mAb) or Sirukumab and/or c) showing values that are comparable to generally accepted developability standards for therapeutic antibodies[45]. None of the selected antibodies exhibit any polyspecificity signature (e.g., binding curvature during association phase and/or slower off-rate during dissociation) in contrast to polyspecific control mAb (binds ssDNA and cardiolipin) and CC12.17 (binds ssDNA). As expected, the extent of self-interaction as measured via AC-SINS shows a clear positive wavelength shift (greater values indicative of increased self-reactivity) for Sirukumab ($10.1 \pm 2.6$ nm), which is in line

with the worst 10% value reported previously ($11.8 \pm 6.8$ nm)[45]. In contrast, selected mAbs exhibit minimal shift, with the worst antibody among the selected ones (F7) exhibiting a minimal wavelength shift ($0.59 \pm 0.08$ nm) that slightly outperformed clinically approved Abrilimab ($0.74 \pm 0.22$ nm) (Supplementary Fig. 7a). All tested mAbs show more favorable Tm values (increasing values indicative of more thermally stable) than poorly developable Sirukumab (>$68.5 \pm 0.5$ °C), with the worst antibody performer among the selected antibodies (G12) exhibiting a value of $69.3 \pm 1.5$ °C, comparable to the parental ($70.0 \pm 1$ °C). The mean $\pm$ standard deviation across all selected

**Table 2 Neutralization potencies of the selected antibodies obtained against the SARS-CoV-2 virus.**

| Clone | Live SARS-CoV-2 virus IC50 (ng/mL) |
|---|---|
| #41 | 3.4 |
| A12 | 1.3 |
| B3 | 5.6 |
| D2 | 34.8 |
| D3 | 1.8 |
| F1 | 1.7 |
| F7 | 3.5 |
| G2 | 3.3 |
| G5 | 12.1 |
| G12 | 1.7 |

**Table 3 Neutralization potencies of the selected antibodies obtained against the Alpha and Beta variants.**

| Clone | SARS-CoV-2 pseudovirus B.1.1.7 (Alpha variant) IC50 (ng/mL) | SARS-CoV-2 pseudovirus B.1.135 (Beta variant) IC50 (ng/mL) |
|---|---|---|
| #41 | 2.8 | 4.3 |
| A12 | 2.9 | 1.0 |
| B3 | 5.1 | 6.5 |
| D2 | 34.7 | 10.4 |
| D3 | 3.2 | 1.8 |
| F1 | 5.2 | 3.3 |
| F7 | 6.7 | 3.7 |
| G2 | 4.2 | 1.8 |
| G5 | 70.5 | 30.4 |
| G12 | 6.2 | 3.5 |
| CC6.29 | 42.8 | 528.1 |
| CC6.30 | 9.0 | >1000 |
| Ab1 | 154.0 | >1000 |
| STE73-2E9 | 7.7 | >1000 |

antibodies was $72.9 \pm 4.5\,°C$. It should be emphasized that Sirukumab, while exhibiting the least thermally stable value in this set of experiments, is still considered thermally stable in this assay if one considers recently established thresholds ($\geq 64.5\,°C$)[45]. All selected antibodies exhibit low aggregation propensity (lower values indicative of lower aggregation propensity) relative to Sirukumab ($0.23 \pm 0.16\%$), with the worst antibody performer among the selected antibodies (F7) exhibiting a % aggregation/day value of $0.05 \pm 0.03\%$, with a mean ± standard deviation across the set of selected antibodies of $0.02 \pm 0.03\%$ (Supplementary Fig. 7c), all less than the threshold previously established value (over 30 days) of $0.08 \pm 0.03\%$[45].

## Discussion

One silver lining to the emergence of the COVID-19 pandemic has been the intense focus of numerous institutions and scientific actors to solving problems related to the pandemic. This has provided a unique global opportunity to directly compare different technologies applied to the same problem. Perhaps most striking are the vaccine development efforts. Presently, 103 vaccines against 32 different organisms (including strains) are approved for use in the US (https://www.fda.gov/vaccines-blood-biologics/vaccines/vaccines-licensed-use-united-states). However, for only a few (e.g., Herpes Zoster, Typhoid, Influenza) is more than one type of vaccine available (e.g. attenuated virus and recombinant protein); for most, there is just a single vaccine type, reflecting years of research. The advent of COVID-19 completely

upended this slow development paradigm. In just over a year, 354 vaccines (194 preclinical, 135 clinical, and 25 approved), comprising eight different vaccine classes (protein subunit, replicating and nonviral vectors, DNA, RNA, live and attenuated virus, and virus-like particles) have been developed against this single organism (https://www.who.int/publications/m/item/draft-landscape-of-covid-19-candidate-vaccines). Although the outcome of this vaccine comparison is still pending, complicated by the challenge of analyzing the results of immunizing large cohorts in which infection rates and viral variants are a moving target, the RNA vaccines were the fastest to approval, and so far, appear to have the fewest side-effects. Another area of intense focus has been in the sphere of diagnostics, with over 500 different commercial tests available based on four main technologies (PCR, isothermal amplification and antigen or antibody detection) (https://www.360dx.com/coronavirus-test-tracker-launched-covid-19-tests), each of which has different advantages and applications. For example, PCR tests are the most sensitive, but may be too sensitive if positive in the absence of infectious virions.

The comparison of different methods to generate antibodies against the spike protein of SARS-CoV-2, and in particular, its RBD domain[8–23,25–31,48–56], may be more straightforward, in that in vitro antibody properties can be compared to one another using relatively few parameters: affinity and biological activity, which in the case of anti-spike antibodies can be represented by IC50, and the neutralization of variants of concern. Naïve antibody display libraries were first described in 1991, 30 years ago[24,57], with the promise they would replace immunization as the means by which antibodies would be generated in the future. Although many different libraries have been described (e.g.[58–67]), and antibodies from some have entered the clinic and gained approval, phage antibodies have been described as having less favorable developability properties[45] and lower affinities, and therefore routinely require subsequent maturation to develop utility[68]. However, it has been challenging to compare the properties of antibodies derived from different naïve or immune platforms, since they are frequently tested on different targets and are isolated by disparate approaches. The COVID-19 pandemic has provided the unique opportunity to carry out such a direct comparison, with multiple groups worldwide applying their platforms to generating antibodies against a single discrete target: the ~220 amino acids of the CoV-2 Spike protein receptor-binding domain (RBD). Most studies have used B-cells from immune sources, particularly convalescent COVID-19 patients[9–23,48,50,69–78], yielding many antibodies with potent neutralization properties, including some that have received Emergency Use Authorization. However, all RBD antibodies derived from immune sources are not equally potent, with best affinities spanning over four orders of magnitude, and only seven publications reporting best affinities <100 pM (Fig. 5)[10,12,21,48,72,74,76], and a couple of exceptional antibodies with single-digit pM affinity[48,74]. In the case of naïve libraries, there is a three order of magnitude span of best reported affinities[25–32,79], with only one publication[79] reporting a single antibody with an affinity below 100 pM. IC50s for live virus neutralization show a similar trend, with the eleven best published immune antibodies having IC50s < 10 ng/ml[13,15,17,18,21,48,50,71,77,80], and no published naïve best antibody having a live virus IC50 < 10 ng/ml.

Here, within the context of this international antibody comparison, we show that a semi-synthetic antibody library, relying on whole natural CDRs for diversity, can rapidly generate many antibodies with affinities and neutralization activities against CoV-2 that are superior to those previously reported from naïve libraries and comparable to the best from immune convalescent patients. Figure 5 compares the affinities and IC50s for the ten most potent antibodies selected here (in each category) of the 23 tested, against the single most potent antibody from every

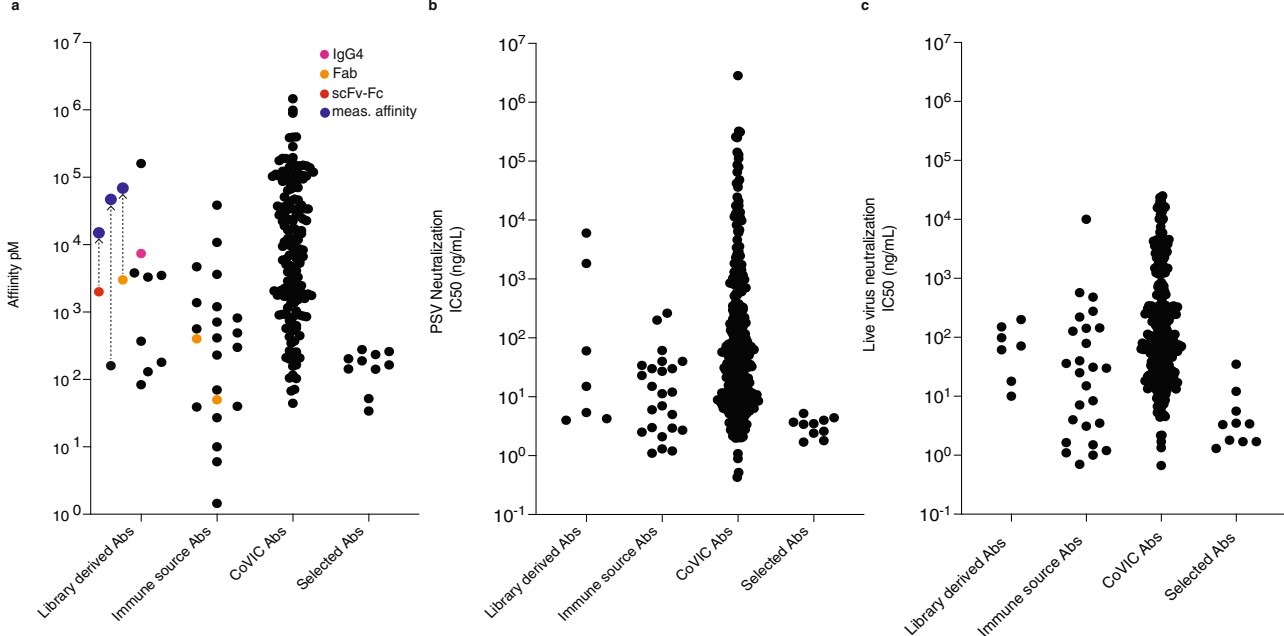

**Fig. 5 Comparison of selected SARS-CoV-2 spike antibodies to the published record. a** For each of the individual publications (see introduction) the IgG, Fab, or scFv-Fc with the highest affinity for the RBD is indicated (derived either from recombinant naïve antibody libraries "Library derived Abs" or from immunized animals or recovered Covid19 patients "Immune source Abs"), together with the affinities of 181 antibodies in the CoVIC Ab database recognzing the RBD ("CoVIC Abs"). For the antibodies selected here ("Selected Abs"), the ten IgGs with the highest affinities for the RBD are indicated. For three of the naïve library derived antibodies we showed the reported values and those obtained here ("Meas. Affinity"). **b** For each of the individual publications the IgGs with the lowest $IC_{50}$'s obtained using pseudovirus are indicated, while for the CoVIC database the $IC_{50}$'s of 306 reported antibodies are indicated For the antibodies selected here, the 10 IgGs with the lowest pseudovirus $IC_{50}$s are indicated. **c** For each of the individual publications the IgGs with the lowest $IC_{50}$'s obtained using live virus are indicated, while for the CoVIC database the $IC_{50}$'s of 288 antibodies assessed by UTMB are indicated (76 antibodies had $IC_{50}$'s > 25 µg/ml and are not included). For the antibodies selected here, the ten IgGs with the lowest livevirus $IC_{50}$s are shown. For the publications as well as the antibodies selected here, the top antibody in each individual category is not necessarily the same antibody across all categories (e.g., a high-affinity antibody with poor $IC_{50}$'s is reported in 5a, but not 5b or 5c). Furthermore, affinity, PSV, and live virus $IC_{50}$s were not measured in all publications, so only the best measurements in each category that were reported are indicated. Source data are provided as a Source Data File.

published report[9–23,25–28] in the same categories. As can be seen, the range of antibodies described here is generally comparable to, if not better than, the published record of the best antibodies generated by immunization from most publications, with the highest affinity antibody described here (34 pM) having a higher affinity than 79% of the best immune antibodies described, and all the antibodies from naïve libraries. Notably, of the 23 antibodies we tested, 19 possessed subnanomolar affinities, with two having binding affinities below 100 pM. One caveat to this comparison is that affinities in different publications were measured using different approaches in different laboratories (and we found some affinities to be lower when remeasured—see Fig. 5). Fortunately, a recent publication[81] from the Coronavirus Immunotherapy Consortium (CoVIC) compared over 350 mAbs from 56 different labs for affinity and neutralization activity. The antibodies were derived from COVID-19 survivors, phage display, naïve libraries, in silico methods and other strategies, but were submitted blinded to CoVIC, making it impossible to define the origins (naïve or immune) of individual ones. Affinities were assessed using a single technique (surface plasmon resonance) in a single lab, on a single platform (Carterra LSA), which is the same as that used here. 181 of these antibodies recognized the RBD, and comparing their affinities to those described here, shows that the latter have higher affinities than the vast majority of those tested by CoVIC when measured in the same way on the same equipment. Eight of ten tested antibodies had $IC_{50}$s < 10 ng/ml in live virus neutralization assays (and 13 of 23 in pseudoviral assays), including four with $IC_{50}$s < 2 ng/ml and one with an $IC_{50}$ of 1.3 ng/ml,

values significantly better than all the best naïve antibodies, most of the best immune antibodies and all but a few of the CoVIC antibodies.

Although our selections were conducted using the entire S1 domain of SARS-CoV-2, all the high-affinity antibodies we describe here recognized targets in the RBD-A region. This may reflect either a higher immunogenicity or increased stability and folding of the RBD compared to the rest of S1, which may make in vitro antibody selection more effective.

The selected antibodies were compared with a series of previously characterized antibodies. Five were obtained from convalescent patients, two chosen for having high neutralization activities (CC6.29 and CC6.30), two because they bound a different epitopic region (CC12.17 and CC12.18)[18], and one that was cross-reactive with SARS-CoV-1 and recognized an additional, epitope[39]. We also compared four antibodies derived from in vitro "naïve" phage library selections (naïve antibodies) with the highest reported affinities for which sequences were available[39–41]. The affinities of the antibodies from convalescent patients, as produced and measured here, were similar to those previously reported[7,18]. This was not the case for the antibodies from the naïve libraries, which had affinities significantly lower than those previously reported[39–41], a finding we are unable to explain. For all but a few antibodies, values improved ~sixfold when RBD affinities were compared to a trimeric form of the Spike antigen (Fig. 2), reflecting the expected avidity effect due to multivalent binding. Interestingly, some antibodies (e.g., C7, G5, and A12) showed lower or similar avidities for the trimer

compared to the monomeric RBD, which may be related to different trimer conformations[82], allosteric effects (e.g., masking of recognized epitopes in the native trimer), or the fact that the soluble S trimers we used contained an artificial trimerization sequence, which may not conform with that of the natural membrane-bound trimer. It is intriguing that the affinity of the most potent neutralizing antibody (A12 - IC50 1.3 ng/ml) falls within this group, with affinities roughly equivalent for the RBD (52 pM) and trimer (83 pM).

The great majority of the selected antibodies binned together, targeting the RBD-A epitope, and overlapping with sites recognized by CC6.29/30 and most other neutralizing immune antibodies. Most of the tested antibodies were able to inhibit the B.1.17 and B.1.351 variants with IC$_{50}$s similar to those of the wild type (Fig. 4d), indicating their potential utility in antibody cocktails designed to control variants and avoid escape mutants. Two of the selected antibodies (#41 and B3) belonged to a separate binning group, able to block binding to both RBD-A (CC6.29 and CC6.30) and one of the RBD-B antibodies (CC12.18)[18], but these possessed reduced efficacy against the B.1.17 and B.1.351 variants (Fig. 4d).

In conclusion, the studies enabled by this global pandemic comparison show that carefully designed and constructed recombinant naïve antibody libraries are as efficient as immune sources for the generation of highly potent antibodies against a viral pathogen. This establishes that naïve antibody libraries have finally come of age, fulfilling their original promise of bypassing the need to identify immune sources with strong responses to generate powerful antibodies[24]. This library design has shown value not just for infectious disease, but also for therapeutic antibody discovery in general, having produced antibodies with comparable affinities and developability properties against other therapeutic targets[33,34]. The absence of developability concerns in the selected antibodies is manifested by the presence of few identifiable sequence liabilities (Supplementary Fig. 2), mean and median hydropathy and charge profiles for the merged CDRs better than the anti-Covid antibodies in the Cov-AbDab database (Supplementary Fig. 3), a lack of polyreactivity (Supplementary Fig. 4) and values for AC-SINS, Tm and accelerated stability considered developable within the clinical-stage antibody landscape[45]. These features are expected to accelerate scale up production and clinical approval of antibodies selected from this platform.

## Methods

**Phage-display selection**. ScFv antibody fragments were first selected by phage display from a semi-synthetic library[34] using a previously described protocol[35,36]. Briefly, the selections were performed with the automated Kingfisher magnetic bead system (ThermoFisher) using 50 nM of biotinylated SARS-CoV-2 S1 protein (S1N-C82E8, ACRO Biosytems) incubated with the phage antibody library, and capturing binding phage using streptavidin-conjugated magnetic beads (Dynabeads M-280). Non-binding phage are removed from the beads after a series of washing steps and the remaining phage are recovered from the beads by acid elution and used to infect F′ pilus carrying bacteria (Ominmax-2T1, Thermo Fisher Scientific). After propagation of the eluted phage the selection cycle was reiterated, using two rounds of selection.

**Phage ELISA**. Phage ELISA was performed to determine the binding activity of the phage after two rounds of selection before subcloning into the yeast display vector, pDNL6[35,36]. ELISA wells (Nunc Maxisorb) were coated with 0.5 µg neutravidin per well overnight at 4 °C. The following day, excess neutravidin was washed away, and 0.2 µg per well of biotinylated SARS-CoV-2 RBD (ACRO Biosytems, SPD-C52H3), biotinylated SARS-CoV-2 S1 protein (ACRO Biosytems, S1N-C82E8) or an unrelated biotinylated antigen were added. Phage were diluted 1:10 in phosphate-buffered saline (PBS) containing 5% (w/v) skim milk and incubated in the antigen-coated and control wells for 1 h at room temperature. After a rinse step, anti-M13-horseradish peroxidase (HRP) (SinoBiological, 11973-MM05T-H) diluted 1:5000 in PBS and incubated for 1 h at room temperature was used to detect the binding phage. After a final rinse step, 3,3′,5,5′-Tetramethylbenzidine (TMB) was used to

develop the colorimetric assay, which was stopped with 1 M H$_2$SO$_4$. The signal was measured at 450 nm absorbance.

**Yeast display and sorting of scFvs**. After confirming specific binding of the selected phage by ELISA, the second-round phage output was PCR amplified with specific primers (All primer sequences are listed in Supplementary Data 1) that introduce overlapping regions with the yeast display vector pDNL6, allowing cloning by homologous recombination after co-transforming the vector and the amplification products into competent yeast cells[35,36]. The transformed yeast cells were enriched for S1-specific binders by applying three rounds of flow cytometry sorting (FACSAria, Becton Dickinson), following previously described protocols[35,36]. After induction, $2 \times 10^6$ yeast cells were incubated with 50 nM of biotinylated SARS-CoV-2 S1 protein (first enrichment step), 10 nM (second enrichment step), and 1 nM (third enrichment step). Cells were labeled with streptavidin-Alexa-Fluor 633 (1:400) to detect binding of biotinylated target antigens and 0.5 µg/mL of anti-SV5-PE (ThermoFisher #37-750) to assess scFv display levels. Yeast clones showing both antigen binding (Alexa-Fluor 633 positives) and display (PE positives) were sorted.

**IgG reformatting expression and purification**. The strategy is illustrated in Supplementary Fig. 1. Briefly, to maintain the original VL-VH pairing, plasmid DNA purified from the yeast display selection output was amplified by inverse PCR using pools of 3′ primers annealing to the framework 4 (FW4) of the VL region and 5′ primers annealing to the FW1 of the VH region (Supplementary Data 1). This results in the creation of a long plasmid-sized amplicon, in which VH and VL maintain their pairing, but without the scFv linker between VL and VH regions. The PCR product carries overhangs complementary to overhangs on the 3′ and 5′ primers used to PCR amplify the LC constant regions and the HC control elements from two donor vectors, pDONOR_Kappa and pDONOR_Lambda (Supplementary Fig. 1a and Supplementary Data 1). The donor fragment encodes the: (1) CL-kappa or CL-lambda of human IgG1; (2) translation stop site; (3) polyadenylation site; and (4) an SV40 promoter and signal sequence for the heavy chain. Kappa and lambda reactions were performed separately to minimize cross-priming, using a proof-reading polymerase. The donor fragment was then cloned into the linearized vector by homologous assembly using the NEBuilder® HiFi DNA assembly cloning kit (NEB). The resulting intermediate pool consists of antibody cassettes with complete IgG1 LC and a partial HC region, lacking the LC control elements and the HC constant domain (Supplementary Fig. 1b, c). Plasmid DNA was then purified from the intermediate pool and the antibody cassette was obtained by enzymatic restriction digestion. The resulting VL-cassette-VH construct was cloned into the final H chain constant domain vector, which provided the (1) signal peptide and (2) promoter for LC expression; and (3) the HC constant region domains (CH1, hinge, CH2, and CH3) (Supplementary Fig. 1c–e).

**Antibody expression and purification**. Antibody expression was performed using the Expi293 Expression System (Thermo Fisher A14527), according to the manufacturer's protocol. Briefly, Expi293F cells were cultured in Expi293 Expression Medium in a humidified 8% CO$_2$ incubator at 37 °C, on an orbital shaker platform. On the day prior to transfection cells were subcultured to $2 \times 10^6$ viable cells. Expifectamine 293 transfection reagent and 1 µg plasmid DNA/mL of transfection culture volume were separately diluted in OptiMEM I Reduced Serum Medium. Following a 10 min incubation, the Expifectamine 293 mixture was combined with the DNA dilution incubated for 30 min, and the final mixture added to cells. Enhancer 1 and Enhancer 2 were added to transfected cultures 18 h post-transfection and incubated for 3 days. Following harvest, NaN$_3$ was added to supernatants, centrifuged at $3200 \times g$ for 30 min in a refrigerated centrifuge and filtered through a 0.22-µm filter. Recombinant Protein A sepharose (Novex) was added to the supernatants and rotated for 18 h at 4 °C. Antibody was eluted from PBS-washed Protein A beads with 0.2 M glycine-HCl (pH 2.5). Antibody-containing fractions were neutralized, concentrated, and dialyzed in excess PBS prior to final characterization by ELISA and SDS-PAGE.

**Developability assays**. All assays were performed as previously described[45], with modifications from the original assays described in the sections below.

**Polyspecificity with SPR**. All kinetics conditions were carried out as outlined in "Kinetics" section below. The polyspecificity reagents used in this assay include those detailed previously[43], with adjusted concentrations of 50 µg/mL, 4 µg/mL and 10 µg/mL for cardiolipin (C0563; Sigma), ssDNA (D8899; Sigma), and LPS (tlrl-eblps;InvivoGen), respectively.

**Size-Exclusion Chromatography after 15-day exposure to 37 °C**. Samples were kept at 200 µg/mL at 37 °C for 15 days in 1xPBS with 5 µg injected for analysis. All control (day-0) samples were maintained at −20 °C for the duration of the experiment. For Size-Exclusion (SEC) analysis, the running buffer was 50 mM phosphate and 150 mM sodium chloride, pH 7.0. The percent aggregated, measured on the SEC, was determined over the 15-day time course relative to 0-day timepoint and reported as %aggregation/day.

**Informatics Processing**. The Cov-AbDab (Raybould et al.[38]) was downloaded on June 29th pertaining to the 02/28/21 version of the database. A total of 1,643 (non-redundant) full-length sequences classified as SARS-CoV-2 in addition to 31 characterized antibodies from this study were processed together using IMGT annotation (KABAT for LCDR2) to quantify sequence-based liabilities and bio-physical properties including net charge, length, and Parker hydropathy[83].

**ACE-competition ELISA**. Greiner Microlon 200 plates (Thermo Fisher Scientific) were coated with 50 ng/well of recombinant RBD in 0.1 M bicarbonate buffer pH 9.6 and incubated overnight at 4 °C. Wells were blocked with 2% MPBS (1x PBS, 2% skim milk (w/ v)) for 30 min at 37 °C. After washing, serial dilutions of purified antibodies were added to the wells in the presence of 0.5 μg/mL of biotinylated ACE2 protein and incubated for 1 h at 37 °C. After washing, the bound ACE2 was detected by adding AP-conjugated streptavidin (0.6 mg/mL) diluted 1:2000 in PBS and incubating for 1 h at 37 °C. After the final wash step, the immunocomplexes were detected by adding 1 mg/ml of p-nitrophenol/DEA buffer at pH 9.8 and reading the plates at 405 nm.

**Kinetic measurements**. For the kinetics experiments an HC200M sensor chip (Carterra #4297) was activated with 33 mM of N-Hydroxysulfosuccinimide (S-NHS, sigma #56485)), 133 mM N-(3-Dimethylaminopropyl)-N'-ethylcarbodiimide hydrochloride (EDC, sigma #E7750) diluted in 0.1 M of MES, pH 5.5 for 5 min. An antihuman Fc (Southern Biotech, #2048-01) was diluted to 50 μg/mL in 10 mM Acetate, pH 4.33, and immobilized on the chip for 20 min. The chip surface was deactivated using 1 M ethanolamine, pH 8.5, to prevent any additional primary amine coupling. Individual antibody clones were diluted to 10 μg/mL in 1x HBSTE (Carterra #3630) and printed onto the chip for 12 min. Antigen was prepared across a 7-point dilution series from 100 nM to 137 pM (SARS-CoV-2-Trimer, SPN-C52H9 ACRO Biosystem) or 300 nM to 411 pM (RBD) in 1xHBSTE supplemented with 0.5 mg/mL BSA. Association was set to 5 min, with a dissociation time of 11 min per cycle. Binding kinetics were fit using the Kinetics software suite (Carterra) using a one-site binding model.

**Binning**. For the binning experiments an HC30M sensor chip (Carterra #4279) was activated with 33 mM of N-Hydroxysulfosuccinimide (S-NHS, sigma #56485)), 133mM N-(3-Dimethylaminopropyl)-N'-ethylcarbodiimide hydrochloride (EDC, sigma #E7750) diluted in 0.1 M of MES, pH 5.5 for 5 min. All ligand antibodies (panel of antibodies to be tested) were diluted in 10 mM NaAcetate, pH 4.33 with 0.01% tween. All antibodies were coupled at 10 μg/mL in a 96-well plate. Activation with S-NHS/EDC occurred over a 5-minute period with coupling to ligand mAbs at 10 min. To hydrolyze unused S-NHS esters back to original form, we used 50 mM Borate buffer flow for 15 min followed by 1 M ethanolamine wash for 2 × 30 s washes to block any potential non-hydrolyzed sites from 50 mM Borate wash. In scouting experiments, antigen RBD or Trimer were injected at different concentrations to determine the best binding and regeneration conditions. From these studies, we used the Trimer at 50 nM concentration, diluted in 1xHBSTE supplemented with 0.5 mg/mL Bovine Serum Albumine (BSA). Regeneration was optimal when using IgG Elution buffer, glycine, pH 2.8 (ThermoScientific). Analyte mAbs were prepared at 75 μg/mL diluted in running buffer (1xHBSTE supplemented with 0.5 mg/mL BSA). The Trimer protein was diluted to 50 nM for the binning run. We set buffer injections to occur every 12 analyte injections with a panel of ~40 antibodies tested for each run. Injection times were set to 1 min baseline, 4 min for antigen injection, 4 min for mAb analyte, 1 min dissociation, and 2 × 20 s regeneration step determined by the scouting experiments. All analysis was conducted using the Epitope software (Carterra). All thresholds were set globally or adjusted to level to account for self-blocking of the ligand(self)/analyte/mAb (self) combination.

**Pseudovirus neutralization assay**. Pseudovirus (PSV) preparation and assay were performed as previously described with minor modifications[18]. Pseudovirions were generated by co-transfection of HEK293T (ATCC CRL-3216) cells with plasmids encoding (HIV-1) NL4-3 ΔEnv Vpr Luciferase Reporter Vector (pNL4-3.Luc.R-E-) and SARS-CoV-2 spike WT (GenBank: MN908947) or variants (B.1.1.7 variant or Alpha variant; B.1.351 variant or Beta variant) with an 18 aa truncation at the C-terminus. Supernatants containing pseudotyped virus were collected 48 h after transfection and frozen at −80 °C for long-term storage. PSV neutralizing assay was carried out as follows: 25 μl of mAbs serially diluted in DMEM with 10% heat-inactivated FBS, 1% Q-max, and 1% P/S were incubated with 25 μl of SARS-CoV-2 PSV at 37 °C for 1 h in 96-well half-well plate (Corning, 3688). After the incubation, 10,000 Hela-hACE2 cells generated by lentivirus transduction of wild-type Hela (ATCC CCL-2) cells and enriched by fluorescence-activated cell sorting (FACS) using biotinylated SARS-CoV-2 RBD conjugated with streptavidin-Alexa Fluor 647 (Thermo, S32357) were added to the mixture with 20 μg/ml Dextran (Sigma, 93556-1G) for enhanced infectivity. 48 h postincubation, the supernatant was aspirated, and HeLa-hACE2 cells were then lysed in luciferase lysis buffer (25 mM Gly-Gly pH 7.8, 15 mM MgSO4, 4 mM EGTA, 1% Triton X-100). Bright-Glo (Promega, PR-E2620) was added to the mixture following the manufacturer's instruction, and luciferase expression was read using a luminometer. Antibody

samples were tested in duplicate, and assays were repeated at least twice for confirmation. Neutralization ID$_{50}$ titers or IC$_{50}$ values were calculated using "One-Site Fit LogIC50" regression in GraphPad Prism 9.

$$100 * \left( 1 - \frac{RLUs\ of\ Sample - Average\ RLUs\ of\ Background}{Average\ RLUs\ of\ Virus\ Control - Average\ RLUs\ of\ Background} \right) \quad (1)$$

**Live virus neutralization assay**. Vero E6 cells (ATCC CRL-1586) were seeded in 96-well half-well plates at approximately 8000 cells/well in a total volume of 50 μl complete DMEM medium (DMEM, supplemented with 10% heat-inactivated serum, 1% GlutaMAX, 1% P/S) the day prior to the in-cell ELISA. The virus (500 plaque-forming units/well) and antibodies were mixed, incubated for 30 min and added to the cells. The transduced cells were incubated at 37 °C for 24 h. Each treatment was tested in duplicate. The medium was removed and disposed of appropriately. Cells were fixed by immersing the plate into 4% formaldehyde for 1 h before washing 3 times with PBS. The plate was then either stored at 4 °C or gently shaken for 30 min with 100 μL/well of permeabilization buffer (consisting of PBS with 1% Triton-X). All solutions were removed, then 100 μl of 3% BSA was added, followed by room temperature incubation at 2 h.

The antibodies selected against the spike protein primary antibodies were diluted in PBS/1% BSA to a final concentration of 2 μg/ml. The blocking solution was removed and washed thoroughly with wash buffer (PBS with 0.1% Tween-20). The primary antibody mixture, 50 μl/well, was incubated with the cells for 2 h at room temperature. The plates were washed 3 times with wash buffer.

Secondary antibody (Jackson ImmunoResearch, Peroxidase AffiniPure Goat Anti-Human IgG (H + L), #109-035-088) diluted to 0.5 mg/ml in PBS/1% BSA was added at 50 μL/well and incubated for 2 h at room temperature. The plates were washed 6 times with wash buffer. HRP substrate (Roche, Cat. No. 11582950001) was freshly prepared as follows: Solution A was added to Solution B in a 100:1 ratio and stirred for 15 min at room temperature. The substrate was added at 50 μL/well and chemiluminescence was measured in a microplate luminescence reader (BioTek, Synergy 2).

**Data analysis**. The following method was used to calculate the percentage neutralization of SARS-CoV-2. First, we plotted a standard curve of serially diluted virus (3000, 1000, 333, 111, 37, 12, 4, 1 PFU) versus RLU using four-parameter logistic regression (GraphPad Prism ver. 8) below:

$$y = a + \frac{b - a}{1 + \left( \frac{x}{x_0} \right)^c} \quad (2)$$

(y: RLU, x: PFU, a, b, c, and x$_0$ are parameters fitted by standard curve)

To convert sample RLU into PFU, use the equation below: (if y < a then x = 0)

$$x = x_0 \log_c \frac{b - y}{y - a} \quad (3)$$

Percentage neutralization was calculated by the following equation:

$$\%Neut = 100 \times \frac{VC - nAb}{VC - CC} \quad (4)$$

VC = Average of vehicle-treated control; CC = Average of cell-only control, nAb, neutralizing antibody. PFU value was used for each variable indicated.

To compute neutralization IC$_{50}$, logistic regression (sigmoidal) curves were fit using GraphPad Prism. Means and standard deviations are displayed in the curve fit graphs and were also calculated using GraphPad Prism.

**Reporting Summary**. Further information on research design is available in the Nature Research Reporting Summary linked to this article.

## Data availability

The sequences of the light and heavy chain variable regions of the selected antibodies can be found in Supplementary Table 1 and on GenBank (https://www.ncbi.nlm.nih.gov/genbank/) under the accession numbers MZ927183, MZ927184, MZ927185, MZ927186, MZ927187, MZ927188, MZ927189, MZ927190, MZ927191, MZ927192, MZ927193, MZ927194, MZ927195, MZ927196, MZ927197, MZ927198, MZ927199, MZ927200, MZ927201, MZ927202, MZ927203, MZ927204, MZ927205, MZ927206, MZ927207, MZ927208, MZ927209, MZ927210, MZ927211, MZ927212, MZ927213, MZ927214, MZ927215, MZ927216, MZ927217, MZ927218, MZ927219, MZ927220, MZ927221, MZ927222, MZ927223, MZ927224, MZ927225, MZ927226, MZ927227, and MZ927228. These sequences can also be found on Figshare (https://doi.org/10.6084/m9.figshare.17004013.v1) and Zenodo (https://doi.org/10.5281/zenodo.5686448). For Supplementary Fig. 3, we used the CoV-AbDab (http://opig.stats.ox.ac.uk/webapps/covabdab/). For Fig. 5, we accessed data from the CoVIC-DB database (https://covic.lji.org/). Source data are provided with this paper.

## Material availability

All requests for resources and reagents should be directed to the Corresponding Author (abradbury@specifica.bio). This includes antibodies, plasmids, and proteins. All reagents

will be made available on request after completion of a Material Transfer Agreement. Source data are provided with this paper.

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

## Acknowledgements

This work was partially funded by NIH, R01-AI-113266 from the National Institutes of Health awarded to ARMB.

## Author contributions

F.F., S.D., M.F.E, A.P., D.R.B., A.R.M.B. conceived and designed the study and performed supervision of the project and experiments. F.F. performed the library slelection. W.H., D.C., and C.L.L. cloned and expressed the antibodies. F.F., D.H., A.A.T., M.F.E., performed sequencing, bioinformatic, data handling and analysis: S.D., A.A.T., M.F.E. measured affinities and performed the binning characterization: M.F.E. executed the developability assays. D.H., L.P., J.E.V., D.M., D.C., W.H., A.C. conducted the neutralization assays. F.F., M.F.E., D.R.B., A.P., A.R.M.B wrote and edited the manuscript.

## Competing interests

Fortunato Ferrara, M. Frank Erasmus, Sara D'Angelo and Andrew R.M. Bradbury are employees and stockholders of Specifica Inc. All other authors declare no competing interests. The library format described in this paper is covered by the following patent and patent applications, with Specifica always as patent applicant and Andrew R.M. Bradbury, M. Frank Erasmus and André A. Teixeira as inventors: 10,954,508 application status: granted; 17/208,877 application status: allowed; 17/163,170 application status: pending; number EP 19834483.0; application status: pending; Number JP 2021-523556; application status: pending; Number CA 3106115 application status: pending. The antibodies described in the manuscript are covered by the following patent application: patent applicant: Specifica Inc, inventor names: Andrew Bradbury, Fortunato Ferrara, M. Frank Erasmus, number 63/211432, application status: provisional.
