## [Transparent Peer Review File · Nature Communications]

Peer Review Information

Reviewer comments, first version:

Reviewer #1 (Remarks to the Author: Overall significance):

The SARS-CoV-2 global pandemic has resulted in many research groups attempting to isolate antibodies against SARS-CoV-2 using a wide variety of different methods. In this manuscript, the authors describe the use of their semi-synthetic naïve scFv library to screen for scFvs that bind to the S1 domain of SARS-CoV-2 spike protein. The library used in the study is constructed using their patented platform that combines antibody scaffolds from clinical mAbs with CDRs (except CDRH3) from BCRs sampled from multiple donors that have sequence liabilities removed to improve the developability properties. The CDRH3 diversity is generated via amplification directly from BCR mRNA. Starting with this semi-synthetic library, the authors completed 2 rounds of phage panning before transferring the screened scFv variants into a yeast display system to do 3 additional rounds of FACS sorting. Subsequently, they sequenced 96 clones, which resulted in 31 unique sequences, and they selected 23 to characterize their functional features as full-length IgGs.

Their principal finding is that some of the antibodies they isolated from their naïve scFv library demonstrate extremely high binding affinities (as low as 34 pM to RBD and 38 pM to S trimer) with potent neutralization activities in their lentivirus pseudovirus assay (IC50 of 1.7 ng/mL) as well as live virus assay using Vero cells (IC50 of 1.3 ng/mL). The authors used previously published 5 mAb isolated from B cell profiling of Covid-19 patients as a benchmark, and several of their antibodies demonstrated greater potency than those reference mAbs.

These antibodies represent the best antibodies that have been isolated from naïve libraries, in terms of binding affinities and neutralization activities. Their findings are of great significance that demonstrates the impressive potential of the semi-synthetic library they have constructed, especially given their focus on the developability features of the library.

Reviewer #1 (Remarks to the Author: Impact):

One potential drawback of a synthetic library-based antibody discovery approach is the lower developability of selected antibodies. The authors address this through their generation 3 antibody

library designed to eliminate CDRs with sequence liabilities, and they also demonstrate the antibodies selected from the library having comparable potency as antibodies isolated from B cells of convalescent patients.

Reviewer #1 (Remarks to the Author: Strength of the claims):

The technical approach for mAb characterization, including detailed binding and neutralization assays, is robust and convincing. On the other hand, there are some key concerns/questions, which are further discussed below (as well as in the "Reproducibility" section).

1. Construction of the naïve library:

It would strengthen the manuscript to include additional details regarding the construction of the initial phage library used for screening (some of the information included in the accompanying manuscript).

2. Developability properties:

While the authors have checked for poly-specificity and sequence liabilities of the selected antibodies, but additional measurements into stability/aggregation (e.g., T_m measurement, SEC, etc.) would further strengthen their claim that naïve antibody libraries can effectively compete with B cell profiling to provide therapeutic antibodies without further downstream optimization. Related minor points:

- o Sequencing the 96 clones selected at the end of the screening process resulted in 31 unique sequences. Did the analysis included in Supplementary Figure 2 consider all 31 sequences or the 23 sequences that were selected for further characterization? Figure 2 should include all 31 sequences.
- o Can the authors describe how the 23 antibodies were selected for subsequent biochemical and functional assays? Were all 31 antibodies cloned and tested as IgG but 8 were removed because they didn't convert to the IgG format? The authors should discuss.

3. Control antibodies:

CC6.29, CC6.30 (targeting RBD-A), CC12.17, CC12.18 (targeting RBD-B), and CR3022 (targeting RBD-C) were selected as 5 control antibodies to benchmark the antibodies selected from their library. Based on the data presented in the Rogers et al study where these antibodies were characterized, it appears that they were chosen for having high neutralization activities (except CR3022), although they do not have the highest binding affinities. Authors should discuss their rationales for choosing CC6.29/30 for RBD-A and CC12.17/18 for RBD-B. Also, the authors should consider including a couple of previously published antibodies isolated from naïve libraries in their assays.

4. Figure 5 should not include any binding data from antibodies evaluated in non-IgG format (i.e. Fab and scFv) for consistent and unbiased comparisons.

Reviewer #1 (Remarks to the Author: Reproducibility):

1. Data description (a major concern):

There are numerous inconsistencies regarding how the binding and neutralization data are described in the main texts and presented in Figure 2A/Table 1, which are detailed below. For clarity, the authors should correct any inconsistencies.

o Line 116-7: The avidities of the selected antibodies to trimeric spike range from 38 pM to 32.7 nM (not 37 pM to 32 nM) according to Figure 2A. The median values also appear to be inconsistent with the data in Figure 2A.

o Line 117-8: 399 pM and 242 pM need to be switched.

o Line 118-9: According to Figure 2A, the binding affinities of CC12.17, CC12.18, and CR3022 to RBD are 84.6 nM, 2.2 nM, and 11 nM, respectively, but the texts says "1.6 nM, \geq 50 nM and 11 nM".

o Line 142: According to Table 1, IC50 values ranges from 1.7 to 85.2 ng/mL, not 1.7 to 92 ng/mL.

2. Some of the figures are not legible (e.g., Supplementary Figures 3 and 4). Please edit those figures.

3. Supplementary Figure 6 is not referenced in the manuscript.

4. While Figure 4D nicely shows the ratios of neutralization activities of the selected mAbs against the UK and SA variants, please include the neutralization IC50 values for each.

5. Table 2, please include antibody names next to the neutralization values included in the table.

Reviewer #2 (Remarks to the Author: Overall significance):

Ferrara et al describe the isolation of panel of high affinity and potently neutralizing SARS-CoV2 antibodies from a large library of naive scFvs using a combination of phage and yeast display. The authors report the successful isolation of a relatively large panel of high affinity ($KD < 100$ pM) antibodies some of which were shown to be neutralizing and importantly a few also neutralized VOCs including B.1.351. These are interesting and timely findings and point to the utility of the author's naive library and screening and IgG characterization processes. Having said that what I find objectionable in the paper is the casting of the story as a "competition" for SARS-CoV2 antibodies compared to other antibody discovery efforts. Frankly, this is pointless and a thinly veiled corporate propaganda. It makes little sense to compare different methods for isolation antibodies that were performed by different labs using very different approaches and with different objectives. For many of the studies referenced here the emphasis was not isolating the highest affinity clones to begin with. Furthermore why is the isolation

of high affinity antibodies from a naive library deemed to be more significant (ie "winning" the "competition") than using screening followed by affinity maturation. For example using naive libraries + affinity maturation Kosiakoff and coworkers (ref 41) and others have found low pM affinity, neutralizing antibodies.

Before publication it is critical that the paper is revised to recast the storyline and eliminate any mention of competition and simply present the findings for what they are. Additionally the authors should address the following technical points:

Major:

- 1) Antibody sequences must be shown. Also some analysis and brief discussion of the CDR3s (length, aa composition etc) should be presented.
- 2) The authors argue that the antibodies isolated are not likely to have developability liabilities. However this is based solely on sequence analysis and sidesteps perhaps the most critical issues, propensity to aggregation and stability. The authors should at the very least present SEC data to evaluate propensity for aggregation for at least a few of the IgGs (say n=5-6) as well as some stability data

Minor:

- 1) Table 2: The Table does not have the antibodies for which the EC50 values are reported!
- 2) SI Fig 4: The polyreactivity SPR data have no positive control! Because avidity is important in polyreactivity assays ELISA may be a more relevant (but not required for the paper).

Reviewer #2 (Remarks to the Author: Strength of the claims):

See comments to authors.

Reviewer #2 (Remarks to the Author: Reproducibility):

Need to provide the sequences of the antibodies for reproducibility.

Author rebuttal, first version:

Point-by-Point Response to Reviewers

Reply to Reviewer #1

We are very pleased that Reviewer #1 found our findings of great significance, we also appreciated his/her/their comments and concerns, here are our responses:

1. Construction of the naïve library:

It would strengthen the manuscript to include additional details regarding the construction of the initial phage library used for screening (some of the information included in the accompanying manuscript).

Construction of the antibody library has been described in much greater depth as requested. The manuscript describing the library concept, construction and validation, which accompanied submission of this paper has now been positively reviewed (requiring minor edits), and a reference has been added.

2. Developability properties:

While the authors have checked for poly-specificity and sequence liabilities of the selected antibodies, but additional measurements into stability/aggregation (e.g., T_m measurement, SEC, etc.) would further strengthen their claim that naïve antibody libraries can effectively compete with B cell profiling to provide therapeutic antibodies without further downstream optimization.

Following the reviewer's suggestion, we added additional measurements to validate the developability properties of the selected antibodies, including affinity-capture self-interaction nanoparticle spectroscopy (AC-SINS), size-exclusion chromatography (SEC) after accelerated stability assay (exposure to high temperature, 37°C) and T_m measurement. The selected antibodies described in the study all derived from a single scaffold obtained from a clinical antibody, which was also tested in parallel in those assays, together with Sirukumab a poorly developable antibody that failed in advanced stage of clinical trials and was used as a positive control for poor developability properties. Furthermore, the number of sequence liabilities present in the selected antibodies were compared to those found in the CoV-AbDab of SARS-CoV-2 spike protein antibodies.

3. Sequencing the 96 clones selected at the end of the screening process resulted in 31 unique sequences. Did the analysis included in Supplementary Figure 2 consider all 31 sequences or the 23 sequences that were selected for further characterization? Figure 2 should include all 31 sequences.

The reviewer is right that the liability analysis was performed on the 23 antibodies selected for further characterization. We modified the Supplementary Figure 2, adding the analysis of all 31 sequences compared to a large dataset of antibodies selected against SARS-CoV-2-spike protein (CoV-AbDab).

4. Can the authors describe how the 23 antibodies were selected for subsequent biochemical and functional assays? Were all 31 antibodies cloned and tested as IgG but 8 were removed because they didn't convert to the IgG format? The authors should discuss.

The reviewer is correct, we were not able to further characterize those 8 antibodies because they did not yield sufficient protein for the planned experiments. We have better elucidated this point in the manuscript.

3. Control antibodies:

CC6.29, CC6.30 (targeting RBD-A), CC12.17, CC12.18 (targeting RBD-B), and CR3022 (targeting RBD-C) were selected as 5 control antibodies to benchmark the antibodies selected from their library. Based on the data presented in the Rogers et al study where these antibodies were characterized, it appears that they were chosen for having high neutralization activities (except CR3022), although they do not have the highest binding affinities. Authors should discuss their rationales for choosing CC6.29/30 for RBD-A and CC12.17/18 for RBD-B. Also, the authors should consider including a couple of previously published antibodies isolated from naïve libraries in their assays.

The reviewer is correct, the CC6.29 and CC6.30 antibodies were chosen for their high neutralization activities, the CC12.17 and CC12.18 were chosen because they bound to a different epitope to the CC6 antibodies, while CR3022 was chosen for its ability to cross-react with SARS-CoV and also recognizes a different epitope. We also followed the reviewer's suggestion and added 4 antibodies obtained from naïve libraries in our assays.

4. Figure 5 should not include any binding data from antibodies evaluated in non-IgG format (i.e. Fab and scFv) for consistent and unbiased comparisons.

Rather than removing non-IgG format antibodies in Figure 5, we indicated those antibodies that were in non-IgG formats, using different symbols. Also, we indicated both the published affinities of the naïve antibodies we tested, as well as the affinities we measured.

Reviewer #1 (Remarks to the Author: Reproducibility):

1. Data description (a major concern):

There are numerous inconsistencies regarding how the binding and neutralization data are described in the main texts and presented in Figure 2A/Table 1, which are detailed below. For clarity, the authors should correct any inconsistencies.

- o Line 116-7: The avidities of the selected antibodies to trimeric spike range from 38 pM to 32.7 nM (not 37 pM to 32 nM) according to Figure 2A. The median values also appear to be inconsistent with the data in Figure 2A.
- o Line 117-8: 399 pM and 242 pM need to be switched.
- o Line 118-9: According to Figure 2A, the binding affinities of CC12.17, CC12.18, and CR3022 to RBD are 84.6 nM, 2.2 nM, and 11 nM, respectively, but the texts says “1.6 nM, \geq 50 nM and 11 nM”.
- o Line 142: According to Table 1, IC50 values ranges from 1.7 to 85.2 ng/mL, not 1.7 to 92 ng/mL.

We are sorry we missed those discrepancies before the submission of the manuscript. The data in the figure and in the table are the correct ones and we fixed the text accordingly.

2. Some of the figures are not legible (e.g., Supplementary Figures 3 and 4). Please edit those figures.

We are sorry for the poor quality of the pictures. The new figures should be more legible.

3. Supplementary Figure 6 is not referenced in the manuscript.

Supplementary Figure 6 (now Supplementary Figure 1) explain our sorting strategy. We have corrected the text to cite it correctly.

4. While Figure 4D nicely shows the ratios of neutralization activities of the selected mAbs against the UK and SA variants, please include the neutralization IC50 values for each.

We recalculated all the neutralization activities for all the 10 selected antibodies and all the other antibodies included in the study, including those from convalescent patients as well as naive recombinant antibody libraries. Figure 4 panel D has been updated and in Table 3 we report the neutralization values against the two variants, while in Table 1 the ones against the wild type virus are reported.

5. Table 2, please include antibody names next to the neutralization values included in the table.

We added the antibody names in the Table 2.

Reviewer #2 (Remarks to the Author: Overall significance):

Ferrara et al describe the isolation of panel of high affinity and potently neutralizing SARS-CoV2 antibodies from a large library of naive scFvs using a combination of phage and yeast display. The authors report the successful isolation of a relatively large panel of high affinity (KD<100 pM) antibodies some of which were shown to be neutralizing and importantly a few also neutralized VOCs including B.1.351. These are interesting and timely findings and point to the utility of the author's naive library and screening and IgG characterization processes. Having said that what I find objectionable in the paper is the casting of the story as a "competition" for SARS-CoV2 antibodies compared to other antibody discovery efforts. Frankly, this is pointless and a thinly veiled corporate propaganda. It makes little sense to compare different methods for isolation antibodies that were performed by different labs using very different approaches and with different objectives. For many of the studies referenced here the emphasis was not isolating the highest affinity clones to begin with. Furthermore why is the isolation of high affinity antibodies from a naive library deemed to be more significant (ie "winning" the "competition") than using screening followed by affinity maturation. For example using naive libraries + affinity maturation Kossiakov and coworkers (ref 41) and others have found low pM affinity, neutralizing antibodies.

Before publication it is critical that the paper is revised to recast the storyline and eliminate any mention of competition and simply present the findings for what they are. Additionally the authors should address the following technical points:

As requested by the reviewer we have rewritten the manuscript and discussed the comparison of the properties of antibodies generated using different methods, rather than using the word competition.

Major:

1) Antibody sequences must be shown. Also some analysis and brief discussion of the CDR3s (length, aa composition etc) should be presented.

We added the sequences of the antibodies in Supplementary Table 2 and we describe the CDR3s and the rest of the CDRs in Supplementary Figure 3.

2) The authors argue that the antibodies isolated are not likely to have developability liabilities. However this is based solely on sequence analysis and sidesteps perhaps the most critical issues, propensity to aggregation and stability. The authors should at the very least present SEC data to evaluate propensity for aggregation for at least a few of the IgGs (say n=5-6) as well as some stability data.

We thank the reviewer for this suggestion, and have performed developability assays (polyspecificity, AC-SINS, SEC/Aggregation and Tm determination) for all 10 antibodies we deeply validated for their neutralization properties plus all the other published antibodies used in this manuscript for comparison.

Minor:

1) Table 2: The Table does not have the antibodies for which the EC50 values are reported!

We added the antibody names in the Table 2.

2) SI Fig 4: The polyreactivity SPR data have no positive control! Because avidity is important in polyreactivity assays ELISA may be a more relevant (but not required for the paper).

We agree with the reviewer that ELISA maybe be more relevant for testing polyreactivity, however we found an antibody highly cross-reactive that we were able to use in the SPR assay, showing non-specific binding for ssDNA, cardiolipin and LPS.

Reviewer #2 (Remarks to the Author: Strength of the claims):

See comments to authors.

Reviewer #2 (Remarks to the Author: Reproducibility):

Need to provide the sequences of the antibodies for reproducibility.

We added Supplementary Table 2 with all the sequences.

Reviewer comments, second version:

Reviewer #1 (Remarks to the Author: Strength of the claims):

1. Construction of the naïve library: The revised details regarding the library concept, construction, and validation, along with the ref. 43 significantly strengthens the manuscript.

2. Developability properties: Supp Fig 7 have fully addressed my concerns, and I appreciate the authors performing all of the additional experiments suggested.

3. Figure 5: I appreciate the authors' efforts in addressing my concern. It is unclear to me why >10-fold worse binding affinities are observed for the mAbs indicated as "Meas. affinity" (when compared to the reported values), but this does not affect the claims that are supported by the data.

Overall, all the edits and additional experiments have strengthened the manuscript greatly.

Reviewer #1 (Remarks to the Author: Reproducibility):

I appreciate all the careful edits (especially regarding the data inconsistencies), and all of my concerns have been addressed in the revised version of the manuscript.

Reviewer #2 (Remarks to the Author: Overall significance):

The authors have addressed all my comments in a satisfactory manner.